# The Effect of Mineralocorticoid Receptor 3 Antagonists on Anti-Inflammatory and Anti-Fatty Acid Transport Profile in Patients with Heart Failure

**DOI:** 10.3390/cells11081264

**Published:** 2022-04-08

**Authors:** Xiaoran Fu, Cristina Almenglo, Ángel Luis Fernandez, José Manuel Martínez-Cereijo, Diego Iglesias-Alvarez, Darío Duran-Muñoz, Tomás García-Caballero, Jose Ramón Gonzalez-Juanatey, Moises Rodriguez-Mañero, Sonia Eiras

**Affiliations:** 1Translational Cardiology Group, Health Research Institute, 15706 Santiago de Compostela, Spain; fideliafu@outlook.com; 2Cardiology Group, Health Research Institute, University Hospital of Santiago de Compostela, 15706 Santiago de Compostela, Spain; cris.almenglo@gmail.com (C.A.); iglesias.alvarez.d@gmail.com (D.I.-A.); jose.ramon.gonzalez.juanatey@sergas.es (J.R.G.-J.); 3Heart Surgery Department, University Hospital of Santiago de Compostela, 15706 Santiago de Compostela, Spain; angelluis.fernandez@usc.es (Á.L.F.); jose.manuel.martinez.cereijo@sergas.es (J.M.M.-C.); dario.duran.munoz@sergas.es (D.D.-M.); 4CIBERCV Madrid, Department of Morphological Sciences, University of Santiago de Compostela, 15706 Santiago de Compostela, Spain; 5Morphological Sciences Department, Medicine Faculty, University of Santiago de Compostela and Pathology Department, University Hospital of Santiago de Compostela, 15706 Santiago de Compostela, Spain; tomas.garcia-caballero@usc.es; 6Cardiology Department, University Hospital of Santiago de Compostela, 15706 Santiago de Compostela, Spain

**Keywords:** epicardial fat, parasympathetic dysfunction

## Abstract

Epicardial fat thickness is associated with cardiovascular disease. Mineralocorticoid receptor antagonist (MRA), a pharmaceutical treatment for CVD, was found to have an effect on adipose tissue. Our aim was to analyse the main epicardial fat genesis and inflammation-involved cell markers and their regulation by risk factors and MRA. We included blood and epicardial or subcutaneous fat (EAT or SAT) from 71 patients undergoing heart surgery and blood from 66 patients with heart failure. Cell types (transcripts or proteins) were analysed by real-time polymerase chain reaction or immunohistochemistry. Plasma proteins were analysed by Luminex technology or enzyme-linked immunoassay. Our results showed an upregulation of fatty acid transporter levels after aldosterone-induced genesis. The MRA intake was the main factor associated with lower levels in epicardial fat. On the contrary, MRA upregulated the levels and its secretion of the anti-inflammatory marker intelectin 1 and reduced the proliferation of epicardial fibroblasts. Our results have shown the local MRA intake effect on fatty acid transporters and anti-inflammatory marker levels and the proliferation rate on epicardial fat fibroblasts. They suggest the role of MRA on epicardial fat genesis and remodelling in patients with cardiovascular disease. Translational perspective: the knowledge of epicardial fat genesis and its modulation by drugs might be useful for improving the treatments of cardiovascular disease.

## 1. Introduction

Some authors have already described the association between epicardial fat thickness or volume and cardiovascular disease [1,2,3,4]. Although the adipogenesis ability is lower in epicardial than in subcutaneous stromal cells [5], the increment of epicardial fat has a higher impact on cardiovascular disease [6]. Last year, a computed tomography angiography and transcriptomic analysis of epicardial fat tissue demonstrated a fat attenuation index as an indicator of inflammation, vascularity and fibrosis [7], which is associated with the presence of high-risk plaques [8]. In fact, the same amount of tissue might reflect a different cellular remodelling and adipogenesis dysfunction [9]. The expansion of adipose tissue starts with a differentiation of preadipocytes, which can be inhibited by collagen type I [10] or inflammatory cells [11]. The inability to store triglycerides also contributes to metabolic dysfunction and cardiovascular disease [12]. Some of the described molecules as regulators of adipocytes’ genesis are CD36 and preadipocyte factor 1 (Pref-1). While CD36 is considered a marker of human adipose progenitors [13], with an important role on triglyceride accumulation, Pref-1 [14] is a negative regulator and a preadipocyte marker [15]. The main compounds of stromal vascular cells on adipose tissue are preadipocytes, macrophages and endothelial cells [16]. CD68 and CD31 are one of the macrophages [17] and endothelial [18] markers, respectively. Coronary artery disease (CAD) is associated with a higher macrophage polarization in epicardial fat [19] that could improve the angiogenesis and vascular density [20]. All these types of cells express fatty-acid-binding protein 4 (FABP4) [21,22] that is involved in fatty acid transporters and is upregulated in mature adipocytes. The epicardial fat-FABP4 is associated with atherosclerosis [23] and its plasma levels with cardiovascular death [24]. In addition, a higher infiltration of inflammatory cells (lymphocytes CD3 positive) and pro-inflammatory macrophages in epicardial than subcutaneous fat is also associated with CAD [19,25]. The inflammatory cells can develop a fibrotic remodelling in adipose tissue which is also associated with cardiovascular disease [26]. Thus, collagen type I, alpha 2 (COL1A2), fibroblasts and adipocytes´ progenitor marker, or α-smooth muscle actin (α-SMA) andactivated fibroblast marker, might define the fibrosis stage on epicardial fat. Defensins alpha (DEFA) are the antimicrobial peptides of neutrophils [27] which can be infiltrated into inflamed adipose tissue through the chemokine receptor CXCR2 [28], and adhered to adipocytes, through the integrin CD11b [29]. Thus, neutrophils are the most abundant granulocytes that support pro-inflammatory processes in adipose tissue from obesity [30]. One of the anti-inflammatory adipokines mainly expressed and secreted by epicardial fat is the intelectin-1 (ITLN-1), named also omentin [31,32], that could counteract these inflammatory processes and reduce the cardiovascular disease risk [33].

The known studies suggest a differential cell composition between epicardial and subcutaneous fat from patients with cardiovascular disease which could be regulated by risk factors or treatments, such as mineralocorticoid receptor antagonists (MRA), involved in adipogenesis and fibrosis modulation. Since both mechanisms participate in the cardiovascular disease progression, the knowledge of markers involved in cell types, epicardial adipocyte genesis and their main regulators might improve the therapeutical management of these patients. 

## 2. Material and Methods

### 2.1. Human Open-Heart Surgery Samples

Epicardial adipose tissue (EAT) of right ventricle, subcutaneous adipose tissue (SAT) and/or blood were obtained from 71 patients who underwent open-heart surgery. The exclusion criteria were previous heart surgery or severe infectious diseases. All patients signed informed consent. The Galician Clinical Research Ethics Committee approved the study protocol, which was carried out in accordance with the Declaration of Helsinki. Before extra-corporeal pulmonary circulation, small fat biopsies or blood were taken and immediately processed or stored at −80 °C until being used. 

### 2.2. Human Heart Failure Samples

We also included 66 consecutively admitted patients at Cardiology Department for acute heart failure (HF), excluding those with pregnancy, severe chronic liver or kidney disease, autoimmune or chronic inflammatory diseases, as was described before [34]. Dual-energy X-ray absorptiometry (DEXA) (Prodigy, General Electric Medical Systems, Madison, WI, USA) and scans, using enCore™ software (platform version 13.6, General Electric Medical Systems, Madison, WI, USA) were used for body fat mass determination. The Galician Clinical Research Ethics Committee approved the study protocol, which was carried out in accordance with the Declaration of Helsinki.

### 2.3. Adipogenesis Induction for 14 Days

After washing the fat pads three times, the stromal vascular cells (SVC) from consecutive SAT and EAT of 17 patients were isolated and cultured following the collagenase digestion protocol [5]. Then, cells were or were not induced to adipogenesis with M199 medium (Lonza Biologics, Porriño, Spain) supplemented with 10% foetal bovine serum (FBS), and the adipogenesis cocktail, composed of 5 μg/mL insulin, 250 nM dexamethasone, 0.5 mM methylisobutylxanthine and 1 μM thiazolidinedione (IDMT). All pharmacological drugs were obtained from Merck Life Science S.L.U. (Madrid, Spain). In dedifferentiated epicardial or subcutaneous adipocytes from 4 patients, we performed an adipogenesis treatment with IMT (insulin, methylisobutylxanthine, thiazolidinedione) supplemented or not with aldosterone (1 µM) and/or mineralocorticoid receptor antagonist (MRA) (spironolactone 5 µM).

### 2.4. Cell-Type RNA Transcripts on Fat Biopsies and Adipogenesis Assay

Biopsies from epicardial and subcutaneous fat or stromal cells with or without adipogenesis induction were lysed with AllPrep DNA/RNA/protein mini kit (Qiagen, Hilden, Germany) and RNA was obtained, following the manufacturer’s protocol. After retro-transcription, using the Maxima First Strand cDNA Synthesis Kit (Thermo Fisher Scientific, Waltham, MA, USA), 2 μL of cDNA was used for gene expression analysis with the following primers: adiponectin (ADIPOQ), fatty acid binding protein 4 (FABP4), CD36, preadipocyte factor 1 (PREF1), collagen 1A2 (COL1A2), CD68, CD31, ACTA (a-SMA), CD3, DEFA3, CD11b, CXCR2, ITLN1 and ACTB (β-actin) and FastStart SYBR Green Master (Roche Diagnostics, Mannheim, Germany). Primers are detailed in Appendix A. These primers were amplified by real-time polymerase chain reaction at 40 cycles (95 °C for 30 s, 60 °C for 60 s and 72 °C for 30 s) in a QuantStudio 3 (Thermo Fisher Scientific, Waltham, MA, USA). The cycle threshold (Ct) values of the genes were normalized by the Ct values of ACTB (ΔCt). The differential expression levels were represented as arbitrary units (a.u.) based on 2-^(ACTB/gene)^ algorithm.

### 2.5. Immunohistochemistry

Samples were immersion fixed in 10% neutral buffered formalin for 24 h and embedded in paraffin routinely. Sections 4 µm-thick were mounted on FLEX IHC microscope slides (Agilent, Carpinteria, CA, USA). After deparaffination and epitope retrieval (for 20 min at 97 °C in EnVision FLEX target retrieval solution at low pH for FABP4 and high pH for the remaining antibodies), immunohistochemistry was automatically performed using an AutostainerLink 48 immunostainer (Agilent). Briefly, the slides were incubated at room temperature in: (1) rabbit polyclonal antibodies to: FABP4 (Cloud-Clon Corp., Houston, TX, USA, at 1:1000 for 30 min), Omentin (Bioss Antibodies, Woburn, MA, USA, at 1:500 for 30 min) or CD36 (Invitrogen, Waltham, MA, USA, at 1:1000 for 30 min) or mouse monoclonal antibody to CD68-clone PGM1 (Agilent, ready to use for 20 min); (2) EnVision^®^+ Dual Link System-HRP (Agilent Technologies, Inc., Santa Clara, CA, USA, dextran polymer conjugated with horseradish peroxidase and affinity-isolated goat anti-mouse and goat anti-rabbit immunoglobulins) (Agilent, K4065) for 20 min; (3) DAB+ substrate-chromogen solution (1 mL of substrate buffer solution containing hydrogen peroxide and 20 µL of 3,3′-diaminobenzidine tetrahydrochloride chromogen solution) (Agilent) for 10 min; and (4) EnVision FLEX hematoxylin (Agilent) for 15 min.

### 2.6. Wound Healing Assay

Epicardial and subcutaneous stromal cells from 4 patients were seeded in 24 multiwell plates until reaching the 90% of confluence. Then, a wound was perpendicularly and longitudinally performed through the well by a plastic pipette tip. Migration and proliferation cells were recorded with (Olympus Provi CM20 Incubation Monitoring System, Shinjuku-ku, Tokio, Japan) for 24 h and underwent aldosterone (1 µM) and/or MRA (spironolactone 10 µM) treatment in M199 medium with or without foetal bovine serum (FBS) 10%. The area at the edge of the lesion was quantified and adjusted with the basal area at different times (3, 6, 12 and 24 h) with the Fiji ImageJ software (v1.53f51, National Institutes of Health, Bethesda, MD, USA, RRID:SCR_002285) [35]. The experiment was performed by duplicating with the first passage of cells. 

### 2.7. Released and Expression ITLN-1 by Epicardial Stromal Cells

Epicardial stromal cells from 4 patients were seeded in 6 multiwell plates until reach the 90% of confluence. Then, cells were washed with saline solution, for removing FBS, and cultured with M199 for 24 h. Afterwards, supernatants were collected and concentrated with Amicon Ultracentrifugal 3K columns (Merck KGaA, Darmstadt, DE, Germany). Released ITLN-1 levels were analysed by enzyme-linked immunoassay (ELISA) and cells were lysed with RLT buffer and RNA was extracted as it was indicated above for ITLN-1 expression levels analysis. 

### 2.8. Blood Analytes Measurements

After centrifuging at 1800× *g* for 10 min, plasma samples were stored at −80 °C until use. A magnetic Luminex test kit (R & D Systems, Minneapolis, MN, USA) was used for analysing FABP4 levels (1:2 dilution). The omentin levels were determined with diluted plasma (1:100) using a commercially available ELISA kit (Cloud-clone corp. (CCC, Wuhan, China)), according to the manufacturers’ protocols. Measurements were performed in duplicate, and the results were represented as means. 

### 2.9. Statistical Analysis

Normal distributions were assessed by Shapiro–Wilk test. Continuous variables were presented as mean ± standard deviation and categorical variables were presented as frequency and percentage. Paired comparisons between control and treatment were determined by paired t test. Differences between patients with respect to treatments or risk factors were determined by unpaired t test. Logistic regression analysis was used for analysing the best predictor associated with FABP4. Statistical significance was defined as *p* < 0.05. All analyses were performed using SPSS v22.0. (Software SPSS Inc.; Chicago, IL, USA).

## 3. Results

### 3.1. Stromal Fat Cell Adipogenesis and Cell-Type RNA Transcripts

Out of 71, 17 fat biopsies were also used for adipogenesis assay. The clinical characteristics of all included patients are described on Table 1. Our results showed that the main differential markers on stromal cells between epicardial and subcutaneous fat were ITLN1 (1.62 ± 0.11 vs. 1.46 ± 0.10, *p* < 0.001), which is higher expressed in epicardial stromal cells and CD36 (1.69 ± 0.10 vs. 1.60 ± 0.06, *p* < 0.01), with higher levels in subcutaneous cells. After adipogenesis induction, the adipocyte marker and fatty-acid transporter levels (adiponectin (ADIPOQ), fatty-acid-binding protein 4 (FABP4)) and CD36) were increased in cells coming from subcutaneous and epicardial fat. The increment was even higher in subcutaneous than epicardial cells (1.28 ± 0.10 in SAT vs. 1.16 ± 0.08 in EAT, *p* < 0.001 for FABP4, 1.27 ± 0.11 in SAT vs. 1.21 ± 0.10 in EAT, *p* < 0.05 for AdipoQ). However, a higher increase in fibroblast marker COL1A2 was detected in epicardial stromal cells after adipogenesis induction in comparison with adipogenised-subcutaneous stromal cells (1.05 ± 0.05 vs. 1.01 ± 0.04, *p* < 0.05) (Figure 1A). A high increment of fatty acid transporter levels after adipogenesis was related to high levels of preadipocytes (Pref-1), macrophages (CD68), epithelial (ITLN1) and endothelial (CD31) markers in the stromal basal fraction (Figure 1B).

### 3.2. Cell-Type RNA-Transcripts in Epicardial and Subcutaneous Fat

We studied a huge population of 71 epicardial and subcutaneous fat biopsies from patients who underwent open-heart surgery. Clinical characteristics are specified in Table 2. Fatty acid transporters (FABP4 and CD36), fibroblasts (COL1A2), endothelial (CD31) and myofibroblast (a-SMA) markers were highly expressed in epicardial and subcutaneous fat pads. The main important differences between epicardial and subcutaneous was regarding epithelial marker (ITLN-1). This transcript was highly expressed in epicardial fat (1.91 ± 0.17 vs. 1.69 ± 0.12 a.u., *p* < 0.001). Other genes with higher expression in epicardial than subcutaneous fat biopsies were the myofibroblast (a-SMA) (1.85 ± 0.05 vs. 1.82 ± 0.06 a.u., *p* < 0.01) and neutrophil markers (DEFA3) (1.71 ± 0.11 vs. 1.66 ± 0.09 a.u., *p* < 0.05). However, the fatty acid transporters (FABP4 and CD36) were higher in subcutaneous than in epicardial fat (2.27 ± 0.10 vs. 2.19 ± 0.10 and 2.01 ± 0.05 vs. 1.96 ± 0.05 a.u., *p* < 0.001, respectively) (Figure 2A). The immunohistochemistry demonstrated that the fatty acid transporters are mainly expressed in adipocytes, with a higher size in SAT than EAT, and ITLN-1 is detected in mesothelial cells of EAT (Figure 2B). The high adipocyte size is concordant with their higher ability to be differentiated with aldosterone (down).

### 3.3. Cell Types RNA-Transcripts on Epicardial and Subcutaneous Fat with MRA

After 15 days of adipogenesis with IMT, an upregulation of FABP4 was detected after aldosterone addition in subcutaneous stromal cells (1.8 ± 0.2 vs. 1.5 ± 0.7 a.u., *p* < 0.01), but not in epicardial stromal cells (Figure 2C). However, in biopsies, our results showed epicardial fat-FABP4, CD36 and ITLN1 differences between patients who were taking or not taking MRA. Thus, epicardial fat biopsies from patients who were taking MRA expressed lower levels of CD36 (1.93 ± 0.03 vs. 1.97 ± 0.05 a.u.; *p* < 0.01) and FABP4 (2.14 ± 0.09 vs. 2.20 ± 0.11). On the contrary, there was a higher level of ITLN1 in epicardial fat from patients who were taking MRA (1.98 ± 0.16 vs. 1.90 ± 0.15, *p* < 0.05). However, it was not detected regarding subcutaneous fat (Figure 3). We did not find other different clinical characteristics between patients who were taking or not taking MRA, except atrial fibrillation (AF) presence (Table 3). However, the logistic regression analysis determined that MRA was the best associated factor with FABP4 (β = −0.070; *p* < 0.05) or CD36 levels (β = −0.042; *p* < 0.01) on EAT. Plasma FABP4 levels had a tendency to decrease in those patients with MRA intake (Figure 4). However, it did not reach statistical significance.

In 66 acute HF patients, the circulating FABP4 at discharge, but not ITLN1, was associated with body fat mass (gr), measured by DEXA (r = 0.59, *p* < 0.0001), (Figure 5A). The clinical characteristics of these patients are described in Table 4. However, the low percentage of patients without MRA treatment did not allow us to visualize differences in total body fat between patients with or without MRA intake.

### 3.4. MRA on Epicardial Stromal Cells and ITLN-1

The released ITLN-1 levels in epicardial fat were incremented in epicardial stromal cells after MRA treatment. A post hoc Bonferroni test indicated that the aldosterone combined with MRA treatment was significantly different than control (8.7 ± 2.4 vs. 3.3 ± 1.2 ng/mL, *p* = 0.018). Similar levels of ITLN-1 were released by epicardial stromal cells after aldosterone or control treatment (Figure 6A). However, the gene expression levels of ITLN-1 in epicardial stromal cells were not regulated by MRA treatment after 24 h treatment (Figure 6A).

Regarding epicardial and subcutaneous stromal cell proliferation in a wound healing assay, our results showed that the proliferation rate was not affected at 3 h, 6 h and 24 h after aldosterone treatment. Meanwhile, it was inhibited after MRA treatment (Figure 6B).

## 4. Discussion

For the first time, our results show a local effect of MRA intake on epicardial fat from patients with cardiovascular disease, specifically on fatty acid transporters and on the anti-inflammatory epithelial marker, ITLN-1. Levels of the fatty acid transporter CD36 were higher in subcutaneous stromal cells than epicardial. This might explain their higher adipogenicity. After subcutaneous or epicardial adipogenesis induction, we observed an upregulation of both fatty acid transporters, CD36 and FABP4. The latter, that experienced the highest increment, was related to higher levels of macrophages, endothelial, epithelial and preadipocyte markers in epicardial stromal cells but not in subcutaneous. Thus, the genesis-involved mechanisms of epicardial fat might differ from those of subcutaneous tissue with a macrophage leadership [36,37]. Although, we did not observe differential expression levels of macrophage markers between EAT and SAT biopsies. Higher levels of fatty acid transporters were detected in subcutaneous fat tissue than epicardial. In contrast, the epithelial marker ITLN-1 was higher in epicardial fat tissue as well as neutrophil markers. This last result might indicate a higher inflammatory cell compound in EAT as was described by other groups [31,38]. We demonstrated that ITLN-1 can be released by epicardial stromal cells and upregulated by MRA intake. It might exert a protective role since its low levels were associated with CAD [39]. Moreover, the MRA enhances its secretion levels which could modulate the macrophage activity on the cardiac system [40]. MRA intake was also associated with a lower level of fatty acid transporters, FABP4 and CD36, in epicardial fat. These results might suggest a lower adipocyte size, since both molecules are increased after adipogenesis induction, or lipid accumulation by other non-adipocyte cells in epicardial fat biopsies. Although FABP4 levels are also detected on plasma and they are associated with fat mass in patients with HF, we did not observe any statistical differences between patients with or without MRA treatment. These results might lead the MRA effect into a modulation of inflammatory state on epicardial fat. However, since most patients in the MRA group were also taking statins, which modulate macrophage polarization and fatty acid accumulation [41,42], a synergic effect of both drugs might improve the regulation of FABP4 and CD36 levels in epicardial fat. Moreover, the MRA effects on the reduction in epicardial stromal cell proliferation and migration rate might suggest a remodelling effect, as was described in cardiac tissue [43] or subcutaneous fat [44]. Although, further mechanistic studies are needed. The epicardial fat volume is associated with cardiovascular disease. Some authors have suggested that the dysfunction of subcutaneous fat involves fatty acid deposition on ectopic fat [45]. This procedure includes adipogenesis of stromal vascular cells. At the beginning, it is a protection of the organ against excessive energy [46]. However, its enlargement contributes to metabolic dysfunction. Our previous results have demonstrated that subcutaneous fat has a higher ability to induce adipogenesis than epicardial stromal vascular cells [5]. The differential transcripts between both fat pads have demonstrated a higher CD36 expression level in subcutaneous than in epicardial stromal cells. Since this is a fatty acid transporter, it might be the main cause of higher adipogenesis induction in subcutaneous cells than epicardial. Because FABP4 was highly expressed after adipogenesis induction, it was used as an adipocyte genesis marker. We performed associations between FABP4 levels after adipogenesis and cell markers in stromal cells. Our results showed that CD36 and Pref-1 were positively correlated with adipocyte genesis (measured by FABP4 levels) in subcutaneous fat cells. Although CD36 was already described as a progenitor of white adipose tissue [13], this is the first time that it was verified in patients with cardiovascular disease. However, the highest epicardial adipogenesis, as was indicated by FABP4 levels, was correlated with FABP4, CD36, CD68 and CD31 levels in epicardial stromal cells. The lack of association between epicardial FABP4 levels, after adipogenesis induction, and the preadipocyte marker Pref-1, might suggest a differential fat accumulation process with respect to subcutaneous cells. Moreover, while IDMT is a well-known adipogenic cocktail for subcutaneous stromal cells, other factors, i.e., atrial natriuretic peptide, might be needed for the epicardial preadipocytes’ differentiation [47]. After adipogenesis induction, there is an increase in Pref-1 and COL1A2 in epicardial cells. It might inhibit faster the adipocytes’ genesis in epicardial cells than in subcutaneous, since Pref-1 is a gatekeeper of adipogenesis [48] and explain its lower differentiation ability and higher fibrosis on this fat tissue. After validating the CD36 and FABP4 levels in epicardial fat biopsies, we observed that they were downregulated in patients who were taking MRA. Since CD36 is incremented after epicardial adipogenesis, as well as FABP4, it might indicate an MRA effect on adipocyte number or lipid accumulation. This hypothesis is based on the described effect of the antimineralocorticoid drospirenone on adipocyte genesis inhibition through transcriptional control [49]. However, since blood cells express CD36 [50], MRA might also modulate the presence of inflammatory cells in epicardial fat. The fact that high fat feeding upregulates CD36, MRA might modulate it and mimic a fasting state [51]. In patients with HF, the circulating FABP4 levels was a good indicator of body fat mass. However, they were not modified in patients with MRA intake. In this sense, plasmatic FABP4 levels were not demonstrated to be a good indicator of MRA effects on epicardial fat from patients with cardiovascular disease. Although our population had no long-term follow-up, several studies have tried to demonstrate the preventive MRA benefits on new-onset atrial fibrillation [52] and progression [53]. Thus, the MRA effects on cell-type RNA transcript composition of epicardial fat might be one of the mechanisms involved in this protection since its association with atrial fibrillation risk is already known [26,54]. In fact, a recent clinical trial was designed for studying the effect of MRA treatment on post-operative atrial fibrillation of patients undergoing open-heart surgery [55]. Perhaps, the short period of time with MRA will be not enough for detecting changes in epicardial fat cells’ composition but it might inhibit the aldosterone produced by the epicardial adipocytes [56]. Further studies will be necessary for confirming this hypothesis.

### Limitations

This is a single-centre study. Primary culture cells from patients take a long period of time. This is the main reason for the low number of included patients. This default did not allow us to study the effect of multiple factors on gene expression levels. Subcutaneous fat was not obtained from all included patients. A long-term follow-up was not performed in the study population. Epicardial fat thickness was not quantified in patients undergoing open-heart surgery or hospitalized patients for HF. The duration of MRA treatment before surgery for most patients was not determined and varied from days to years.

## 5. Conclusions

A differential cell-type RNA transcript and adipogenesis function define the epicardial and subcutaneous fat from patients with cardiovascular disease. While higher adiposity is represented in subcutaneous fat, higher epithelial, fibroblast, and neutrophil, markers are found in epicardial fat. MRA intake might modulate the epicardial adiposity and increase the anti-inflammatory profile outstanding to the low FABP4 and ITLN-1 expression levels, respectively.

## Figures and Tables

**Figure 1 cells-11-01264-f001:**
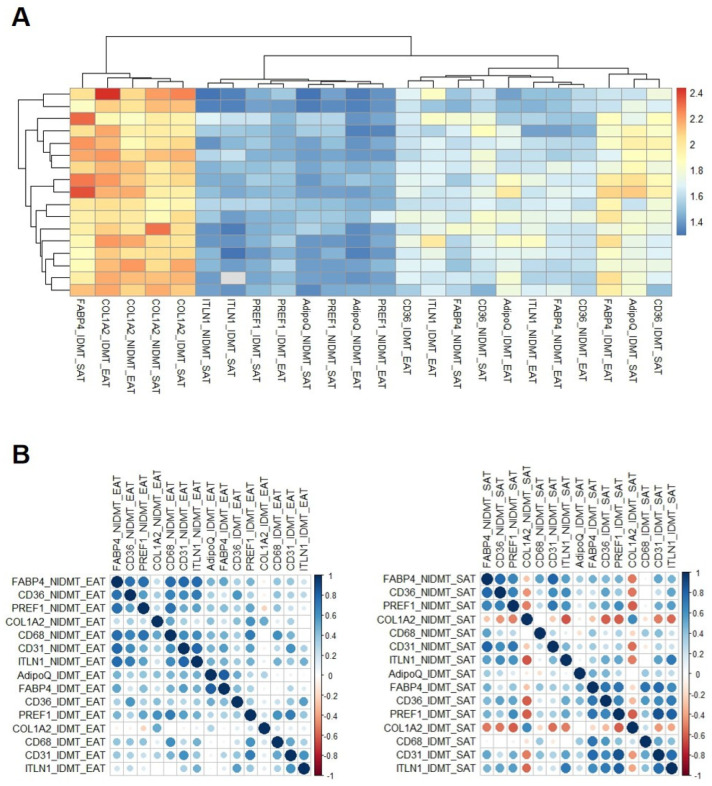
Adipogenesis assays (**A**) Hierarchical clustering heatmap represents the mRNA expression levels of genes on epicardial (EAT) and subcutaneous fat (SAT) stromal cells from patients with cardiovascular disease with or without adipogenesis induction. (**B**) Correlation plots between mRNA expression levels of genes on epicardial (EAT) or subcutaneous fat (SAT) stromal cells from patients with cardiovascular disease with (IDMT) or without adipogenesis induction (NIDMT).

**Figure 2 cells-11-01264-f002:**
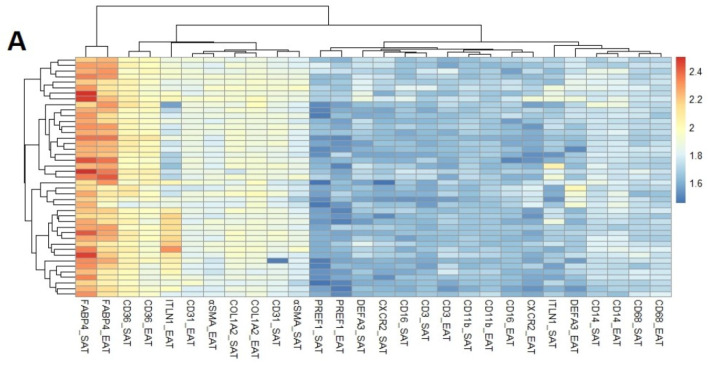
Cell-RNA expression level fat pads (**A**) Hierarchical clustering heatmap represents the mRNA expression levels (a.u.) of genes in epicardial (EAT) and subcutaneous (SAT) fat from patients with cardiovascular disease. (**B**) Immunohistochemistry of EAT and SAT with antibodies against FABP4, ITLN-1, CD36, and CD68 (objective magnification, ×20 and ×40). In numbers, under the immunohistochemistry, the mean ± SD mRNA expression levels of the total analysed biopsies regarding each molecule are represented. (**C**) Dot plots and mean ± SD represent the mRNA expression levels of FABP4 in epicardial and subcutaneous fat stromal cells treated or not treated with aldosterone and mineralocorticoid receptor antagonist (IMT: insulin, IBM and Thiazolidinediones; Aldo: aldosterone; MRA: mineralocorticoid receptor antagonist). Unpaired t test showed statistical differences between groups ** *p* < 0.01.

**Figure 3 cells-11-01264-f003:**
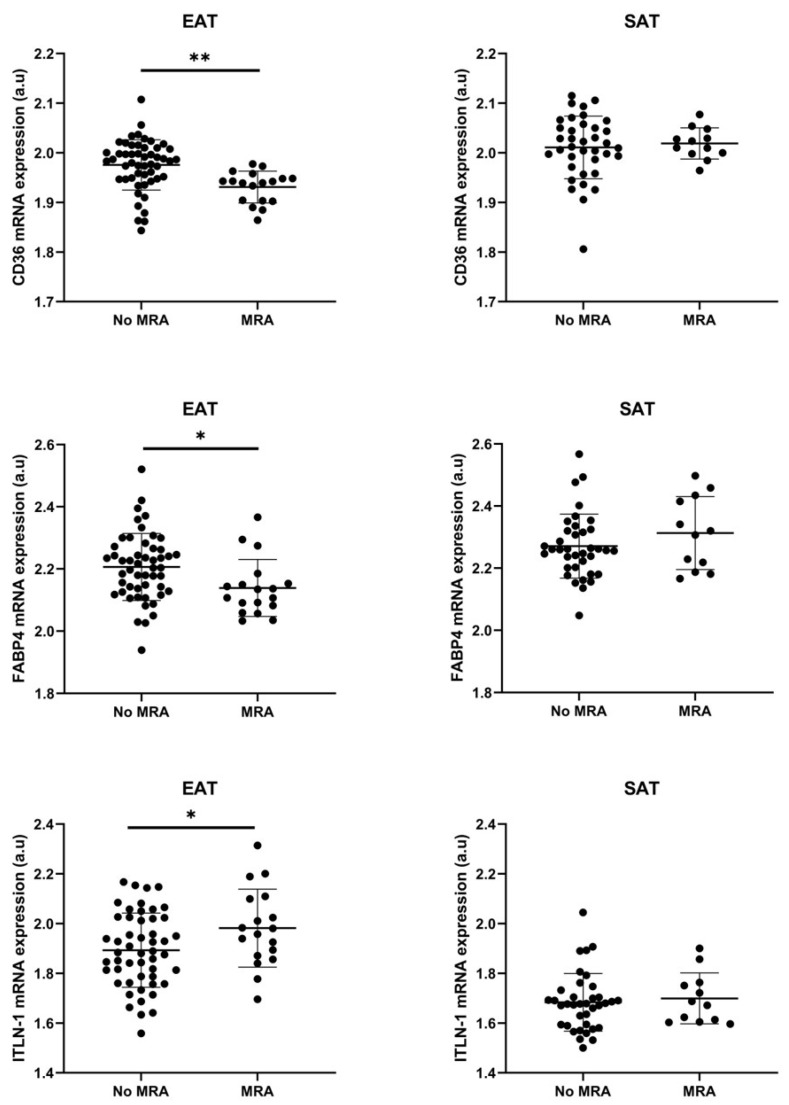
MRA on FABP4, CD36 and ITLN-1 Dot plots and mean ± SD represent the mRNA expression levels of FABP4, CD36 or INTL-1 in epicardial and subcutaneous fat in a.u. from patients with or without mineralocorticoid receptor antagonist (MRA). Unpaired t test showed statistical differences between groups ** *p* < 0.01, * *p* < 0.01.

**Figure 4 cells-11-01264-f004:**
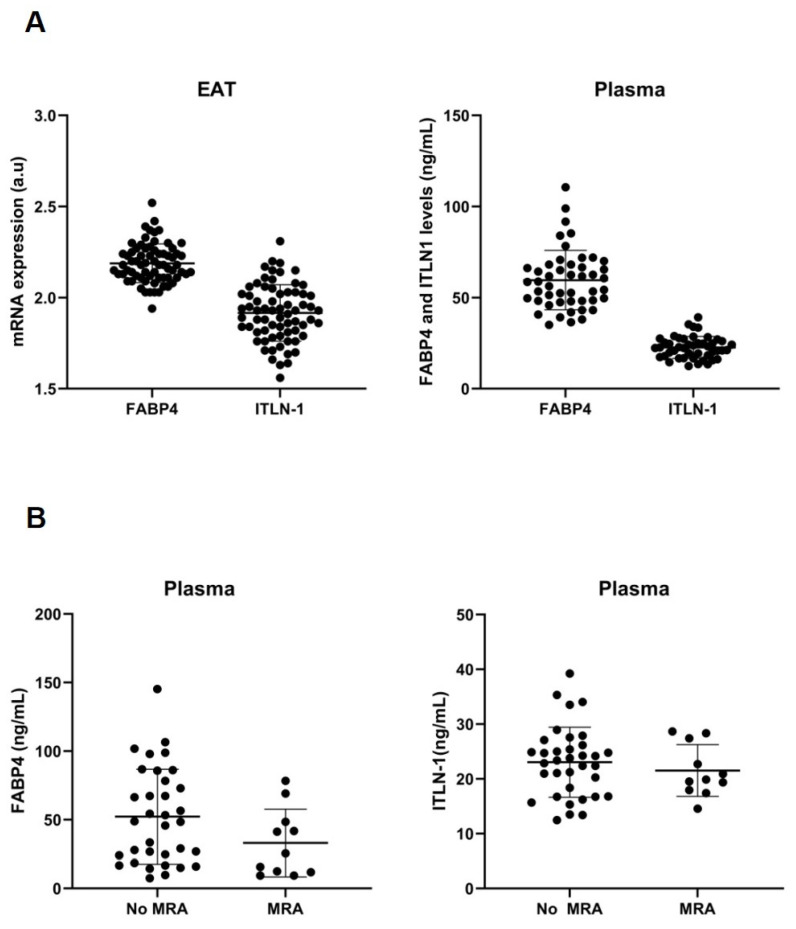
MRA on FABP4, CD36 and ITLN-1 (**A**) Dot plots and mean ± SD represent the mRNA expression levels of FABP4 and ITLN-1 in epicardial fat or plasma levels (ng/mL). (**B**) Dot plots and mean ± SD represent the plasma levels (ng/mL) of FABP4 or ITLN-1 from patients with or without mineralocorticoid receptor antagonist (MRA).

**Figure 5 cells-11-01264-f005:**
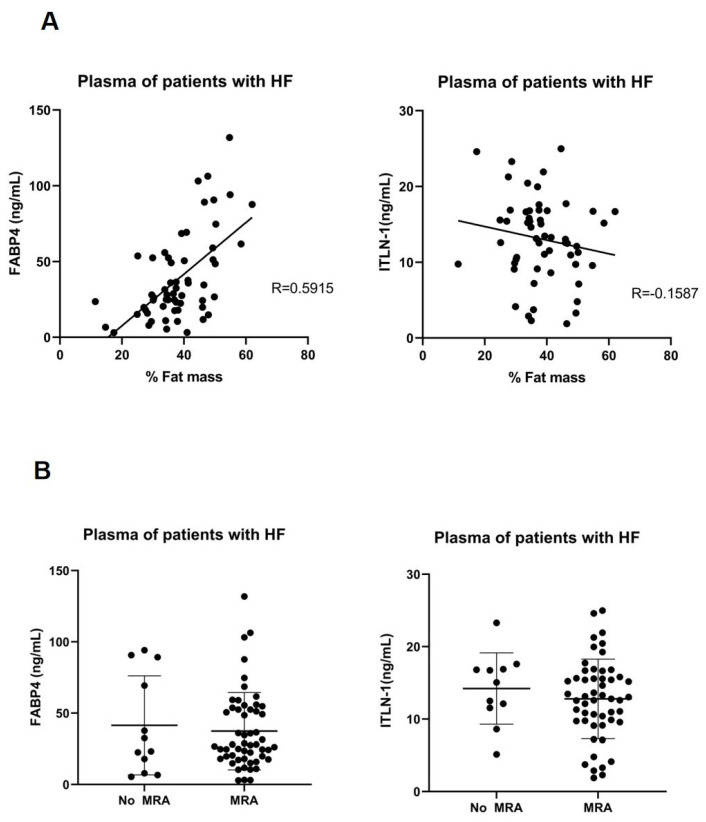
FABP4 and ITLN-1 plasma levels and body fat (**A**) Correlation plot represents the circulating FABP4 or ITLN-1 levels with body fat mass, measured by DEXA, in patients with HF. (**B**) FABP4 or ITLN-1 plasma levels from heart failure patients with or without mineralocorticoid receptor antagonist (MRA). Unpaired t test did not show statistical differences between groups.

**Figure 6 cells-11-01264-f006:**
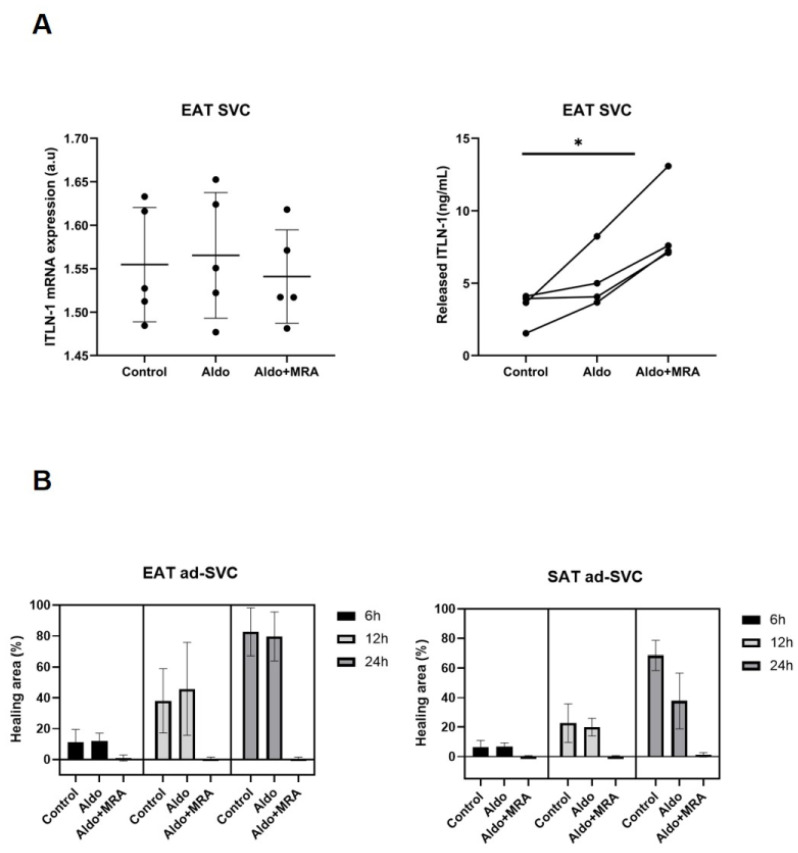
ITLN-1 in epicardial stromal cells and MRA (**A**) Dot plots and mean ± SD represent the released ITLN-1 levels (ng/mL) or mRNA expression levels of ITLN-1 in epicardial fat stromal cells treated or not treated with aldosterone and MRA (**B**) Wound healing assay. Bar plots and mean ± SD represents percentage of healing area of dedifferentiated epicardial and subcutaneous adipocytes (ad-SVC) with or without aldosterone and MRA treatment. ANOVA t test showed statistical differences among groups and Bonferroni post hoc test showed differences between control and MRA treatment * *p* < 0.05.

**Table 1 cells-11-01264-t001:** Clinical characteristics for primary culture adipogenesis assays.

Variables	Mean ± SD	N (Yes/No)	% (Yes/No)
Age (years)	66 ± 8		
Gender (male/female)		14/3	82/18
BMI (kg/m^2^)	29 ± 4		
AHT		9/8	53/47
T2DM		2/15	12/88
Dyslipidemia		9/8	53/47
Smoking		9/8	53/47
HF		8/9	47/53
Valvulopathy		15/2	88/12
CAD		6/11	35/65
AF		7/10	41/59
CRF		1/16	6/94
ACEi		6/11	35/65
b-blockers		10/7	59/41
Statin		9/8	53/47
MRA		3/14	18/82
Oral antidiabetics		2/15	12/88

BMI: body mass index; AHT: arterial hypertension; T2DM: type 2 diabetes mellitus; HF: heart failure; CAD: coronary artery disease; AF: atrial fibrillation; CRF: chronic renal failure; ACEi: angiotensin converting enzyme inhibitor; MRA: mineralocorticoid receptor antagonists.

**Table 2 cells-11-01264-t002:** Clinical characteristics of all patients for biopsies.

Variables	Mean ± SD	N (Yes/No)	% (Yes/No)
Age (years)	70 ± 8		
Gender (male/female)		48/23	32/68
BMI (kg/m^2^)	29 ± 4		
Arterial hypertension		47/24	66/34
T2DM		14/57	20/80
Dyslipidemia		56/15	79/21
Smoking		26/45	37/63
HF		42/29	59/41
Valvulopathy		62/9	87/13
CAD		48/23	34/66
CRF		7/64	10/90
ACEi		19/52	27/73
b-blockers		48/23	68/32
Statin		52/19	73/27
MRA		18/53	25/75
Oral antidiabetics		14/57	20/80

**Table 3 cells-11-01264-t003:** Clinical characteristics of patients with /without MRA.

Variables	MRA	No MRA	*p*
Gender (male/female)	13 (72%)/5 (28%)	35 (66%)/18 (34%)	0.63
AHT (Yes/No)	14 (78%)/4 (22%)	33 (62%)/20 (38%)	0.23
T2DM (Yes/No)	6 (33%)/12 (67%)	8 (15%)/45 (85%)	0.09
Dyslipidemia (Yes/No)	16 (89%)/2 (11%)	40 (75%)/13 (25%)	0.23
Smoking (Yes/No)	7 (39%)/11 (61%)	19 (36%)/34 (64%)	0.82
Heart Failure (Yes/No)	13 (72%)/5 (28%)	29 (55%)/24 (45%)	0.19
Valvulopathy (Yes/No)	17 (94%)/1 (6%)	45 (85%)/8 (15%)	0.29
CAD (Yes/No)	6 (33%)/12 (67%)	18 (34%)/35 (66%)	0.96
CRF (Yes/No)	3 (17%)/15 (83%)	4 (7%)/49 (93%)	0.26
AF (Yes/No)	12 (67%)/6 (33%)	20 (38%)/33 (62%)	0.03
ACEi (Yes/No)	6 (33%)/12 (67%)	13 (25%)/40 (75%)	0.47
b-blockers (Yes/No)	13 (72%)/5 (28%)	35 (66%)/18 (34%)	0.63
Statin (Yes/No)	16 (89%)/2 (11%)	36 (68%)/17 (32%)	0.08
Oral antidiabetics (Yes/No)	6 (33%)/12 (67%)	36 (68%)/17 (32%)	0.09

BMI: body mass index; AHT: arterial hypertension; T2DM: type 2 diabetes mellitus; HF: heart failure; CAD: coronary artery disease; AF: atrial fibrillation; CRF: chronic renal failure; ACEi: angiotensin converting enzyme inhibitor; MRA: mineralocorticoid receptor antagonists.

**Table 4 cells-11-01264-t004:** Clinical characteristics of patients with HF.

Variables	Mean ± SD	N/%	MRA (*n* = 54)	No MRA (*n* = 12)	*p*
Age (years)	71 ± 11	25/62	72 ± 12	71 ± 11	0.80
Gender (male) (*n*/%)			19/35	6/50	0.34
BMI (kg/m^2^)	32 ± 8		31±7	29 ± 6	0,52
AHT (*n*/%)		53/80	44/81	10/83	0.88
T2DM (*n*/%)		33/50	26/48	7/58	0.52
Dyslipidemia (*n*/%)		39/59	29/54	10/83	0.06
Smoking (*n*/%)		16/24	14/26	2/17	0.48
CAD (*n*/%)		15/23	13/24	2/17	0.58
LVEF ≥ 50%		18/27	12/22	6/50	0.05
AF (*n*/%)		2/	2/4	0/0	0.50
ACEi (*n*/%)		43/65	36/67	7/58	0.58
b-blockers (*n*/%)		55/83	46/85	9/75	0.39
Statin (*n*/%)		41/62	33/61	8/67	0.72
Oral antidiabetics (*n*/%)		24/36	17/31	7/58	0.08

## Data Availability

The used and analysed datasets are available from the corresponding author on reasonable request.

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
