# Peer review of "The Effect of Mineralocorticoid Receptor 3 Antagonists on Anti-Inflammatory and Anti-Fatty Acid Transport Profile in Patients with Heart Failure"

_cells, 2022, doi:10.3390/cells11081264_

Round 1

Reviewer 1 Report

Authors investigated properties of epicardial fat in patients undergoing heart surgery and peripheral blood in patients with acute heart failure.

On one hand, sophisticated laboratory methods are used, on the other hand the real impact of results is hard to extract from the article in its current form – for example I was not able to find most of Tables with clinical data. In supplement only Table S1 is available. In this respect, clinical data of patients are to be added and described in more detail in the text - Tables regarding patients characteristics should be shown in the main article not in Supplement – now they are missing even in this section. For example treatment with statins could also change properties of adipose tissue and inflammatory markers in addition to MRA which suddenly appears at the beginning of the Discussion. In addition, other factors could have impact in this and at least should be at least discussed (for example see Kauerova S, et al. Statins Directly Influence the Polarization of Adipose Tissue Macrophages: A Role in Chronic Inflammation. Biomedicines. 2021;9(2):211. doi:  0.3390/biomedicines9020211. Poledne R et al. Polarization of Macrophages in Human Adipose Tissue is Related to the Fatty Acid Spectrum in Membrane Phospholipids. Nutrients. 2019;12(1):8. doi:10.3390/nu12010008.

How some subgroups of patients were selected (17 biopsies from 71 patients)

Some editing of the text could be of value: better explain the meaning of ”deep study, expansion o adipose tissue”, … deviation standard x standard deviation.

In general, main question a main answer in this article should be clearly stated, presented and discussed and based on really available/presented data.   

Author Response

Authors investigated properties of epicardial fat in patients undergoing heart surgery and peripheral blood in patients with acute heart failure. On one hand, sophisticated laboratory methods are used, on the other hand the real impact of results is hard to extract from the article in its current form – for example I was not able to find most of Tables with clinical data.

Response: We are grateful for the reviewer´s comments and we apologize for the unattached tables. There was a mistake in the submission. Now, we have included tables into main manuscript which are really important as the reviewer considered.

In supplement only Table S1 is available. In this respect, clinical data of patients are to be added and described in more detail in the text - Tables regarding patients characteristics should be shown in the main article not in Supplement – now they are missing even in this section.

Response: The mistake was solved. We are sorry for that. The current manuscript has the tables into text, including the drugs intake.

For example, treatment with statins could also change properties of adipose tissue and inflammatory markers in addition to MRA which suddenly appears at the beginning of the Discussion. In addition, other factors could have impact in this and at least should be at least discussed (for example see Kauerova S, et al. Statins Directly Influence the Polarization of Adipose Tissue Macrophages: A Role in Chronic Inflammation. Biomedicines. 2021;9(2):211. doi:  0.3390/biomedicines9020211. Poledne R et al. Polarization of Macrophages in Human Adipose Tissue is Related to the Fatty Acid Spectrum in Membrane Phospholipids. Nutrients. 2019;12(1):8. doi:10.3390/nu12010008.

Response: We are agreeing that statins could also modify the adipose tissue properties, specifically inflammatory markers. We have discussed this point and references were included according journal format.

How some subgroups of patients were selected (17 biopsies from 71 patients).

Response: We have included consecutive patients without considering another parameter. The adipogenesis assays takes long time and the main objective was to define the main associated markers with adipogenesis before and after procedure.

Some editing of the text could be of value: better explain the meaning of ”deep study, expansion o adipose tissue”, … deviation standard x standard deviation.

Response: These typo errors were modified according reviewer´s suggestions.

In general, main question a main answer in this article should be clearly stated, presented and discussed and based on really available/presented data.

Response:  We tried to improve the manuscript with the reviewer´s suggestions. The discussion was modified and some phrases were included into text.

Reviewer 2 Report

  1. The first occurrence of an abbreviation needs to indicate the full term, eg. MRA, EAT, SAT, PCR, CD36, ITLN and etc…. in Abstract or main article.
  2. Please describe more about the candidate markers the authors chose in the introduction of the article.
  3. Page 4, line 162 Table 1; Page 5, line 184, Table 2; Page 8, line 231, Table 3; Page 11, line 248, Table 4. These tables could not be found in the article.
  4. Please describe more about ITLN-1, the significance of ITLN-1 in such patients, and the reasons for choosing ITLN-1 as a study candidate. Please clarify this information in the introduction of the article.
  5. From this study, the author could describe more regarding the genesis-involved mechanism of epicardium might differ from subcutaneous tissues, in the discussion of the revised article.
  6. Could the authors provide the medication use in the enrolled patients? Such as statin therapy, MRA, and etc.
  7. The reason of choosing “parasympathetic dysfunction” as the keyword of the study?
  8. Please include the hypothesis of the study in the introduction of the revised article.
  9. Please discuss more regarding the potential benefit of MRA in reduction of epicardial fat, and its clinical significance
  10. Could the authors provide more information of the enrolled patients, such as LVEF, guidelines direct medical therapy for heart failure and lipid profile.

Author Response

The first occurrence of an abbreviation needs to indicate the full term, eg. MRA, EAT, SAT, PCR, CD36, ITLN and etc…. in Abstract or main article.

Response: We are grateful for the exhaustive revision and comments. We have included the explanation of the abbreviations. We are sorry for this mistake in the previous version.

Please describe more about the candidate markers the authors chose in the introduction of the article.

Response: We have included the description of the candidate markers in the introduction.

Page 4, line 162 Table 1; Page 5, line 184, Table 2; Page 8, line 231, Table 3; Page 11, line 248, Table 4. These tables could not be found in the article.

Response: We are really sorry, something grateful for the exhaustive revision and comments. We have included the explanation of the abbreviations. We are sorry for this mistake in the previous version.

Please describe more about ITLN-1, the significance of ITLN-1 in such patients, and the reasons for choosing ITLN-1 as a study candidate. Please clarify this information in the introduction of the article.

Response: We have included the reason of ITLN-1 analysis in the introduction, according reviewer suggestions.

From this study, the author could describe more regarding the genesis-involved mechanism of epicardium might differ from subcutaneous tissues, in the discussion of the revised article.

Response: The discussion was modified according reviewer´s suggestions.

Could the authors provide the medication use in the enrolled patients? Such as statin therapy, MRA, and etc.

Response: The tables are included into text, we are sorry for the misinformation in the previous version.

The reason of choosing “parasympathetic dysfunction” as the keyword of the study?

Response: We are sorry, this word was a mistake in this manuscript. It is part of our project but not for this article.

Please include the hypothesis of the study in the introduction of the revised article.

Response: The hypothesis was included at the end of the introduction.

Please discuss more regarding the potential benefit of MRA in reduction of epicardial fat, and its clinical significance

Response: The discussion was improved and the protective effects in epicardial fat and the clinical significance.

Could the authors provide more information of the enrolled patients, such as LVEF, guidelines direct medical therapy for heart failure and lipid profile.

Response: We have included LVEF, medical therapy and dyslipidaemia according reviewer suggestions.

Round 2

Reviewer 1 Report

Lot of corrections made.

But still: The main aim (the end of Introduction) should match the begining of Discussion - in abstract this aim is not mentioned - MRA appears in methodology. The Tables are not fully readable including headings and statistical analyses are missing in some (Table 4) - should be improved. 

The Title could be better/more clear - for example: The effect of mineralocorticoid receptor 3 antagonists on anti-inflammatory and anti-fatty acids transport profile in patients with heart failure ... + pls see above

Author Response

We are grateful for the reviewers´s comments and give us the opportunity to improve the manuscript. We have modified the manuscript according the reviewer´s suggestions.

a) Title

b) Aims regarding MRA 

c) Table 4 was improved

Best regards